An ODE model of yaws elimination in Lihir Island, Papua New Guinea

Kimball Presley 1
Levenson Jacob 2
Moore Amy 3
Rychtar Jan rychtarj@vcu.edu 4
Taylor Dewey 4
1 Department of Mathematics, Creighton University , Omaha , NE , United States of America
2 Department of Mathematics, Washington and Lee University , Lexington , VA , United States of America
3 Department of Mathematics and Statistics, Elon University , Elon , NC , United States of America
4 Department of Mathematics and Applied Mathematics, Virginia Commonwealth University , Richmond , VA , United States of America
Ndeffo Martial
Electronic publication date: 2022 Mar 17
Publication date: 2022
Volume: 10
Electronic Location ID: e13018
Received 2021 Oct 18; Accepted 2022 Feb 6
Copyright: ©2022 Kimball et al.
Copyright year: 2022
Copyright holder: Kimball et al.
License: This is an open access article distributed under the terms of the Creative Commons Attribution License, which permits unrestricted use, distribution, reproduction and adaptation in any medium and for any purpose provided that it is properly attributed. For attribution, the original author(s), title, publication source (PeerJ) and either DOI or URL of the article must be cited.
License URL: https://creativecommons.org/licenses/by/4.0/

Keywords: Mathematical model, Yaws, Eradication, Morges strategy, Total community treatment, Total targeted treatment

Funding: The National Security Agency H98230-20-1-0011 The National Science Foundation DMS1950015 Presley Kimball, Jacob Levenson and Amy Moore were supported by the VCU REU program in mathematics funded by the National Security Agency grant number H98230-20-1-0011 and by the National Science Foundation grant number DMS1950015 awarded to Dewey Taylor. The funders had no role in study design, data collection and analysis, decision to publish, or preparation of the manuscript.

==============================
Yaws is a chronic infection that affects mainly the skin, bone and cartilage and spreads mostly between children. The new approval of a medication as treatment in 2012 has revived eradication efforts and now only few known localized foci of infection remain. The World Health Organization strategy mandates an initial round of total community treatment (TCT) with single-dose azithromycin followed either by further TCT or by total targeted treatment (TTT), an active case-finding and treatment of cases and their contacts. We develop the compartmental ODE model of yaws transmission and treatment for these scenarios. We solve for disease-free and endemic equilibria and also perform the stability analysis. We calibrate the model and validate its predictions on the data from Lihir Island in Papua New Guinea. We demonstrate that TTT strategy is efficient in preventing outbreaks but, due to the presence of asymptomatic latent cases, TTT will not eliminate yaws within a reasonable time frame. To achieve the 2030 eradication target, TCT should be applied instead.

Introduction

Yaws is an infectious disease spread by skin to skin contact mostly amongst children (Marks, Solomon & Mabey, 2014). It is caused by bacteria Treponema pallidum ssp. pertenue, and begins at an abrasion or open wound which then develops into the “primary papule” or “mother yaw” (Perine et al., 1984). This initial stage is known as primary yaws and the lesion persists for three to six months (Marks et al., 2015a). A short latency period may occur after primary yaws if the primary papule naturally heals before secondary lesions develop (Perine et al., 1984). Secondary yaws begins with the appearance of other lesions anywhere on the body (Mitjà, Asiedu & Mabey, 2013). These lesions heal spontaneously resulting in a noninfectious latent period that, in some cases, may last the remaining lifetime of the person (Perine et al., 1984). During the latent period, previously infected individuals may relapse into secondary yaws up to 5 years after recovering from infection (Marks et al., 2015b). Up to 10% of yaws cases may develop into late yaws (Mitjà, Asiedu & Mabey, 2013), also known as tertiary yaws. Tertiary lesions tend to be very harmful with massive necrotic tissue destruction; yet they are noninfectious (Marks, Solomon & Mabey, 2014).

Yaws was the first disease to be targeted for eradication by the World Health Organization (Marks et al., 2015a). Before this initiative, 90 countries were reported as endemic, totaling about 50 million cases worldwide (Kazadi et al., 2014). The mass screening and treatment programmes led by WHO reduced the global prevalence by >95% between 1950 and 1964, but yaws has reemerged as a public health problem (Asiedu et al., 2008). In 2018, Papua New Guinea and the Solomon Islands reported over 10,000 suspected cases each (WHO, 2018b). According to the most recent report (WHO, 2020), 15 countries are considered as currently endemic for yaws; 87,877 suspected yaws cases were reported to WHO in 2020 from 11 countries from which 81,369 cases were in Papua New Guinea. Solomon Islands reported 13,694 cases in 2019. Figure 1 shows the most recent status of yaws endemicity.

A single oral dose of azithromycin was shown to be just as effective as the previous treatment of injectable penicillin (Mitjà et al., 2012). This initiated a new wave of interest in eradication of yaws (Marks et al., 2015a) and, in 2012, the WHO implemented the Morges Strategy to combat yaws transmission with the goal of eradicating the disease by 2020 (WHO, 2012). The Morges Strategy includes one or more rounds of total community treatment (TCT) where treatment is given to all members of the community and followed by the total targeted treatment(TTT) where treatment is administered to all actively infected individuals and their close contacts as a response to a local outbreak (WHO, 2012). This strategy continues to be the primary plan to eradicate yaws, but the timeline has been stretched to eradication by 2030 after the original 2020 goal was not met (Dyson et al., 2019).

Figure 1 World map of endemic history and prevalence of yaws.

Data collected from WHO (2018b) and WHO (2020) and map was made with the aid of borders.m file (Greene et al., 2019) in MATLAB.

Mathematical modelling is now a standard and indispensable tool for understanding disease dynamics and control (Anderson & May, 1992), yet there is a surprising lack of models of yaws transmission. Until 2012, the only math model of yaws considered the effect of the chicken pox virus on yaws (Gart & De Vries, 1966). More recently, the model from Mushayabasa et al. (2012) divides the population into rich and poor and preforms a theoretical analysis of disease-free and endemic equilibria. In Muench (2013), the authors fitted a simple catalytic model to age structured yaws data. All other models of yaws are stochastic and were designed to estimate various aspects of yaws eradication. In Fitzpatrick, Asiedu & Jannin (2014), the authors were concerned with the economic side of eradication and concluded that the eradication would not be expensive; yet there is still a large degree of uncertainty for the lack of available data (Dyson et al., 2019). In Dyson et al. (2017), the authors created a model to estimate the fraction of individuals that are missed during treatment in the Morges strategy. In modeling effectiveness of the Morges Strategy, Marks et al. (2017) investigated the probability of eradication. Fitzpatrick et al. (2018) created a linear regression model in order to predict the probability of case reporting in different previously affected countries based on different parameters. Mooring et al. (2019) builds off of the work of Marks et al. (2017), using the same compartment model and modeling the effects of different combinations of TTT and TCT. The most recent model comes from Holmes et al. (2020) in which the authors adapted the model from Dyson et al. (2017) to again simulate different combinations of TTT and TCT. It was found that different populations require different treatments, but in general, TCT was more effective in eradication.

The stochastic models such as recent Marks et al. (2017); Mooring et al. (2019); Holmes et al. (2020) are generally more suitable for the eradication end game than the deterministic compartmental models. Yet, the deterministic models are typically simple and easy to analyze, while still reasonably accurate and realistic. Given the lack of deterministic models of yaws transmission in general, our goal is to develop a deterministic model of yaws transmission and then use the model to compare the effectiveness of TTT and TCT strategies. We use the model to derive a formula for the basic reproduction number and to obtain simulated times needed for yaws elimination. Our model can be used as a quick estimate of the effectiveness of a particular treatment strategy.

Methods

We created a compartmental model shown in Fig. 2. Individuals are born as susceptible (S) at rate Λ. The susceptible individuals become exposed (E) after coming in contact with individuals having primary (Y1) or secondary (Y2) yaws; the transmission rate is β. After an incubation period lasting a time σ−1, the exposed individual develops primary yaws and becomes infectious. The primary yaws lasts a time λ1−1, after which the individual may either develop secondary yaws (Y2) with probability pY1Y2, or go into a first latency period (L1) with probability pY1L1 = 1 − pY1Y2. This means that the rate of progression from Y1 to L1 is pY1L1λ1 while the rate of progression from Y1 to Y2 is pY1Y2λ1.

Figure 2 Scheme of the yaws dynamics.

The arrows denote transitions between the compartments. The letters next to the arrows specify the per capita rates of the transitions. The red arrows denote a treatment. The black arrows show a natural disease progression (without any treatment), from susceptible (S) to exposed (E) and then to primary yaws (Y1). After the primary yaws, a majority of cases goes through a short latent period (L1) before progressing to secondary yaws (Y2). A small portion develops secondary yaws directly. Most secondary yaws heals and the infection becomes latent (L2). The latent cases can relapse to secondary yaws for the rest of their lives. Only a negligible number of individuals develop non-infectious tertiary yaws (Y3).

The average duration of the first latency period is ρ1−1; after that an individual develops secondary yaws. The average duration of secondary yaws is λ2−1. After this period, one could either develop rare, debilitating and very painful, but non-infectious, tertiary yaws (Y3) with probability pY2Y3, or go into the second latent period with probability pY2L2 = 1 − pY2Y3. The average length of the second latency period is ρ2−1. Afterwards, individuals can relapse into secondary yaws with a probability pL2Y2, or develop tertiary yaws with a probability pL2Y3 = 1 − pL2Y2.

Individuals can be treated and return to susceptible at a rate of τI for individuals in a compartment I ∈ {E, Y1, Y2, Y3, L1, L2}. Treatment of each compartment depends on the elimination strategy and the specific values are discussed below.

Finally, all individuals are assumed to die at rate µ.

The parameters are summarized in Table 1. Most parameter values were estimated directly from the literature. The only two exceptions are the transmission rate β and the treatment rates τ. Details are shown in Appendix B.

Table 1 Notation —Parameters, controls and their values.

All times are expressed in months and rates are expressed per month. Details behind model calibration are explained in B.

Symbol	Meaning	Value	Range	Source	
Λ	Birth rate	27.212∗1000	[0.001, 0.003]	United Nations (2019)	
μ −1	Expected life span	65∗12	[600, 1200]	World Bank (2019)	
β	Transmission rate	0.0166	[0.01, 0.02]	Estimated	
σ −1	Length of the incubation period	2130	930,9030	Perine et al. (1984)	
λ 1 −1	Length of primary yaws	3	[3, 6]	Perine et al. (1984)	
λ 2 −1	Length of secondary yaws	3	[0, 60]	Mitjà, Asiedu & Mabey (2013)	
ρ 1 −1	Length of latency after primary yaws	1.5	[1, 2]	Marks et al. (2015a)	
ρ 2 −1	Length of second latency	30	[1, 60]	Perine et al. (1984)	
p Y 1 Y 2	Probability of immediate secondary yaws infection after primary yaws	0.12	[0.09, 0.15]	Mitjà, Asiedu & Mabey (2013)	
p Y 1 L 1	Probability of latency period after primary yaws	1 − pY1Y2			
p Y 2 Y 3	Probability of immediate tertiary yaws infection after secondary yaws	0.0001	[0, 0.0002]	Mitjà, Asiedu & Mabey (2013)	
p Y 2 L 2	Probability of latency period after secondary yaws	1 − pY2Y3			
p L 2 Y 2	Probability of relapsing to secondary yaws during latent period after secondary yaws	0.9999	[0.9998, 1]	Mitjà, Asiedu & Mabey (2013)	
p L 2 Y 3	Probability of developing tertiary yaws during latent period	1 − pL2Y2			
τ I	Rate of treatment for the group I ∈ {E, Y1, Y2, Y3, L1, L2}	variable		See text	

The transmission rate β was obtained by fitting the endemic equilibrium to baseline data (prior the mass treatment, i.e., when all τI = 0) from Lihir Island in Papua New Guinea (Mitjà et al., 2015b).

To model TCT, we assume τI = 1/6 for all I, corresponding to treating the whole population every six months. To model TTT, we assume the best case scenario, i.e., τE = τY1 = τY2 = τY3 = τL1 = 1/6 while τL2 = 0.1/6, i.e., we assume that the TTT strategy finds and treats only 10% of secondary latent cases every six months but otherwise finds and treat every other infected individual. This again corresponds to treating active yaws cases and all their closed contacts (that will be either exposed or at most the first latency period) once in six months. We assume that 90% of secondary latent cases are omitted in the treatment because the were infected independently many months or even years ago and are not close contacts to the currently acutely infected individuals. We adopted these assumptions since we can then demonstrate that even these high coverage, yaws will persist in the population for long time under TTT strategy. The protocol of Mitjà et al. (2015b) study also included a 2 year period of non-strategic treatment. For that period, we assumed τE = τY1 = τY2 = τY3 = τL1 = 1/24 with coverage as in TTT.

We validated our model on data from the mass treatment trial in Lihir Island (Mitjà et al., 2018); see Fig. 3. The fitted curve follows general trends of the data. However, the real data for latent infections exhibits oscillation with peaks and dips every 6 months and our simple model cannot exhibit such oscillations.

Figure 3 Model validation.

Data from Mitjà et al. (2018) (blue circles) track active (left) and latent (right) yaws prevalence on Lihir Island. The study protocol was one round of TCT at time 0, followed by three rounds of TTT every six months till month 24 and then non-strategic treatment till month 42 (Mitjà et al., 2015b). The latent cases data (right) are plotted 3 months earlier to account for the continued seropositivity of latent infections until 3–6 months after treatment (Mitjà et al., 2012). The model predictions show one round of TCT (solid line, months 0–6) followed by TTT (dashed line, months 6–24) and then a period of non-strategic treatment (dotted line, during months 24–42). The black lines represent the model predictions for the parameters as in Table 1. The gray lines represent model predictions when the parameters value vary within the ranges specified in Table 1.

We used the compartmental model from Fig. 2 to create a system of ordinary differential equations. We found disease-free and endemic equilibria. Using the next generation matrix method (van den Driessche & Watmough, 2002), we found the basic reproduction number. We performed the stability analysis of the disease-free equilibria based on methods from van den Driessche & Watmough (2002) and Castillo-Chavez et al. (2002).

We did simulations in MATLAB, the code is made available in supplementary material. We adhered to responsible coding practices as outlined in Lucas et al. (2020).

The global uncertainty and sensitivity analysis by the partial rank correlation coefficients, PRCC was based on Marino et al. (2008). We randomly selected 1000 parameter values within the ranges specified in Table 1. We used only those values that could fit to baseline data from Lihir Island (Mitjà et al., 2015b).

Results

We obtained an explicit formula for the basic reproduction number, R0. As shown in Eq. (9), (1) R0=βσvEvY11+λ1vL2vL1pY1L1ρ1+pY1Y2vL1vL2vY2−pY2L2λ2pL2Y2ρ2

where vI denote the sum of all total rates out of the compartment I, i.e., (2) vE=σ+τE+μ

(3) vY1=λ1+τY1+μ

(4) vY2=λ2+τY2+μ

(5) vY3=μ+τY3

(6) vL1=ρ1+τL1+μ

(7) vL2=ρ2+τL2+μ.

We estimated that without any treatment, R0 = 1.2548. The uncertainty analysis showed that in order to fit data from Lihir Island, R0 is between 1.24 and 1.27; see Fig. 4. With TTT treatment, the values of R0 ranged between 0.02 and 0.08. This shows that TTT is quite effective in prevention of the spreading of the epidemics.

Figure 4 Distribution of R0 for parameters that fit data from Lihir Island (Mitjà et al., 2015b).

Left: Prior the treatment. Right: during TTT treatment.

We proved (Theorem 1 in Appendix A.2) that the disease-free equilibrium is globally asymptotically stable when the basic reproduction number R0 < 1. We also showed (Lemma 1 in Appendix A.2) that R0 is decreasing in the treatment rate. If the treatment rate is high enough, the basic reproduction number drops below 1 even for a conservative TTT strategy when only active cases of yaws gets treated; see Fig. 5. This means that the Morges strategy can eventually eliminate yaws.

Figure 5 Dependence of R0 on τ under TCT and TTT regimes.

Once τ > τ0 where τ0 solves R0 = 1, then the disease can be eliminated. For the parameters as in Table 1, τ0,TCT ≈ 3.5⋅10−4 and τ0,TTT ≈ 1.68⋅10−3.

To understand how long the Morges strategy needs to be applied, we simulated two rounds of initial TCT and followed by subsequent rounds of TTT. We performed a global uncertainty analysis where we varied parameters within the ranges specified in Table 1. Figure 6 demonstrates the results. Our model predicts that it would take about 14 to 16 years to achieve a thousandfold decrease in cases (i.e., less than 1 infected person in Lihir Island). The relatively high prevalence of latent cases in the population and the long latency period are the main culprits behind this long elimination time. The continuous application of TCT strategy every six months can achieve the same results in about 3.5 years; the improvement in speed is caused by the latent cases getting treated as well.

Figure 6 Distribution of times needed to decrease yaws cases thousand times using TTT (left) or TCT (right).

As illustrated in Fig. 7, the success or failure of TTT strategy significantly depends on how many latently infected individuals can be discovered and treated. The figure in fact shows expected elimination times for a whole family of strategies with TTT on one end (when the coverage of L2 is low) and TCT on the other end (when the coverage of L2 is 100%). It can take over 25 years to eliminate yaws if only 1% or less of latent cases are found; it would take about about 10 years if 20% is found and about 5 years if about 50% of cases is found.

Figure 7 Sensitivity analysis of the time needed to eliminate yaws under TTT.

We showed the dependence on the percentage of treated L2 cases explicitly. The analysis of other parameters is done by partial rank correlation coefficients, PRCC (Marino et al., 2008).

The sensitivity analysis shows a strong influence of the relapse rate, ρ2, and the spontaneous healing rate of the secondary yaws, λ2, on the elimination time under the TTT regime; see Fig. 7. The higher the relapse rate and the lower the healing rate, the less it takes to eliminate yaws. This initially counter-intuitive result is caused by the fact that spontaneous healing increases the pool of latently infected individuals that can be missed by the TTT strategy. However, if infected individuals do not heal spontaneously, they can be discovered and treated. This again indicates that the latent individuals are the weakest point of the TTT strategy. A higher birth rate also reduces the time to elimination. This is mainly because a higher birth rate increases the influx of healthy individuals while the active yaws of young children are caught on time before progressing to latency. Naturally, a shorter incubation periods increases the time needed for the elimination as they increase the number of yaws cases. The effects of other parameters are relatively mild and not significant.

Finally, let us note that when R0 > 1, the disease-free equilibrium is not stable and there exists an endemic equilibrium given explicitly in Eq. (10). We run numerical simulations for parameter values with ranges in Table 1 and the numerical solutions of the ODE model always converged to the endemic equilibrium. Moreover, motivated by Yang et al. (2017) and LaSalle (1976), we considered a Lyapunov function L=∑CC−C∗−C∗lnCC∗, where the summation is taken over all compartments C ∈ {S, E, Y1, Y2, Y3, L1, L2} and C∗ is an endemic equilibrium value. It follows that L≥0 and L=0 iff C = C∗ for all compartments. Also, we evaluated L′=∑C1−C∗CC′ at 105 randomly selected values of the compartments. We always saw that L′<0. Thus, we believe that the endemic equilibrium is globally stable whenever R > 1, although we do not have an analytical proof of this fact. However, as it has been shown in Fig. 4, even with the weaker TTT treatment, R0 is significantly less than 1 and thus, for the purpose of the elimination (which is the main focus of this paper), the stability of the disease-free equilibrium is much more important.

Discussion

To model TTT strategy, we made a conservative assumption that not many latent cases are treated. We argue that this is a reasonable reflection of a reality in the eradication endgame. The latent cases represent reservoir of future infections Dyson et al. (2019). By treating a recently relapsed latent case with all its close contacts, TTT strategy prevents outbreaks. However, contact tracing does not identify many other latent cases in the population; they likely got infected independently many months or even years ago. Thus, TTT works quite slowly as an elimination strategy as it is equivalent to waiting for the latent cases to relapse instead of actively identifying and treating them while still asymptomatic.

Our model predicts very little variation of eradication times when using TCT strategy. This is natural as the whole population gets treated and most factors of yaws dynamics thus do not play any crucial role. The variability is much larger for the TTT regime which could potentially eliminate yaws in as little as 14 years but it may also take 16 years. The two key factors responsible for the large variation are the duration of the latent period (which is positively correlated with the elimination time) and the duration of the secondary yaws (which is negatively correlated with the elimination times). Gaining more knowledge about these two parameters would reduce the uncertainty of the model predictions.

Our model differs from previous models in two crucial aspects. First, we developed a deterministic ODE model, in contrast to recent stochastic models developed in Fitzpatrick, Asiedu & Jannin (2014); Marks et al. (2017); Dyson et al. (2017); Fitzpatrick et al. (2018); Holmes et al. (2020). While stochastic simulations can incorporate higher degrees of realism, there is a natural simplicity in the ODE models that allows for an easy estimation of the basic reproduction number. Even with different model parameters, we do not necessarily have to rerun the simulations to be able to predict the model outcomes. Our model can thus serve as a first and reasonably reliable estimate of what will happen under different elimination strategies.

Second, our model incorporates all of the known stages of yaws. All models of yaws should consider susceptible individuals, infectious stage(s) of yaws (possibly divided into primary and secondary yaws) and the asymptomatic latent yaws that can relapse. Similarly to Fitzpatrick, Asiedu & Jannin (2014), we also considered tertiary yaws; and as Mushayabasa et al. (2012), we included exposed individuals. Finally, as in Marks et al. (2017) we included a possibility of a latent period between the primary and secondary yaws.

There are several limitation of our model. Most of the limitations stem from the fact that our model is a simple deterministic ODE model in homogeneous population. It thus cannot capture the true eradication endgame when only very small, often just a single digit, number of individuals are infected. The model also cannot capture household dynamics as done in Dyson et al. (2017) or the population structured into hamlets as done in Mooring et al. (2019).

Unlike in stochastic simulations used in Marks et al. (2017); Dyson et al. (2017); Mooring et al. (2019); Holmes et al. (2020), we do not explicitly consider treatment coverage. An independent coverage is implicitly incorporated in our model—a rate τI = 1/6 can mean that the whole (100%) population is treated once every 6 months, as well as that the attempt to treat the whole population is made every m months but at each attempt, only p∗100% of the population is reached with m/p = 6. A systematic failure of the treatment could be included in the model by duplicating each compartment into “treatment adherent“ and “treatment non-adherent”. Setting the birth rate as (1 − p)Λ and pΛ, respectively, into the susceptible treatment adherent and treatment non-adherent, respectively, would then achieve a systematic failure of treatment for p∗100% of the population.

Economics plays a key role in the feasibility of yaws eradication. Our model should be extended by explicitly optimizing control strategies, i.e., the proper combination of TCT and TTT at the appropriate time intervals. The extensions need to take into the account that underdeveloped areas are more prone to transmission and are harder to screen for active infections.

Conclusions

Our paper is the first ODE compartment model specifically applied to yaws elimination. We investigated two strategies, the total community treatment (TCT) and the total targeted treatment (TTT). In agreement with previous models (Dyson et al., 2017; Marks et al., 2015c), we found that due to the high prevalence of latent infections, it is very hard to eliminates yaws by using TTT. Our model predicts that it would take about 15 years to reduce the prevalence thousandfold from the current levels. On the other hand, it would take only about 3.5 years if the whole community was treated once every six months. This is in a quantitative agreement with a recent detailed stochastic model (Holmes et al., 2020). We also note that due to the global stability of the disease-free equilibrium, and the fact that R0 is significantly less than 1 even under TTT treatment, the initial levels of yaws in the population do not play a crucial role for the eradication.

In the light of above findings, we thus recommend using total community treatment as the primary yaws elimination strategy. This recommendation is further supported by the fact that (a) TCT provides additional benefits such as reduction in trachoma prevalence (Solomon et al., 2015), (b) the cost of TCT is not much larger that the cost of TTT (Fitzpatrick, Asiedu & Jannin, 2014), and (c) TTT requires active surveillance (Fitzpatrick et al., 2018), possibly further erasing the difference between the costs of these two approaches. As a note of caution, our model did not consider emergence of antibiotic resistant strains (Mitjà et al., 2018). It is a question whether a large scale application of TCT could eliminate yaws before the antibiotic resistance becomes a true obstacle.

Supplemental Information

Supplemental Information 1 Matlab code

Click here for additional data file.

Supplemental Information 2 Code to draw a map

Click here for additional data file.

Appendix A. Model analysis

The transmission dynamics described in Section yields the following system of ODEs (8) dSdt=Λ+τEE+τY1Y1+τY2Y2+τY3Y3+τL1L1+τL2L2−βY1+Y2N+μS

(9) dEdt=βY1+Y2NS−σ+τE+μE

(10) dY1dt=σE−λ1+τY1+μY1

(11) dY2dt=pY1Y2λ1Y1+ρ1L1+pL2Y2ρ2L2−λ2+τY2+μY2

(12) dY3dt=pY2Y3λ2Y2+pL2Y3ρ2L2−μ+τY3Y3

(13) dL1dt=pY1L1λ1Y1−ρ1+τL1+μL1

(14) dL2dt=pY2L2λ2Y2−ρ2+τL2+μL2.

A.1 Positivity and boundedness of solutions

All parameters in the model are non-negative and it can be shown that the solutions of the system Eqs. (8)–(14) are non-negative, given non-negative initial values. Indeed, let N = S + E + Y1 + Y2 + Y3 + L1 + L2. The biologically feasible region consists of D⊂R+7 such that N≤Λμ. Adding Eqs. (8) –(14) yields (15) dNdt=Λ−μN

and thus (16) Nt=Λμ−Λμ−N0e−μt.

Consequently, the region D is positively invariant and the model is epidemiologically and mathematically well-posed (Hethcote, 2000). We also see that limn→∞Nt=Λμ.

The system Eqs. (8)–(14) has two equilibria, the disease-free equilibrium and an endemic equilibrium as discussed below.

A.2 Disease-free equilibrium and the basic reproduction number

It follows from Eq. (17) general that the disease-free equilibrium, E0=S0,E0,Y10,Y20,Y30,L10,L20, is given by (17) E0=Λμ,0,0,0,0,0,0.

We calculate the basic reproduction number, R0, using the next generation method (van den Driessche & Watmough, 2002). Let the column vector I = (E, Y1, Y2, Y3, L1, L2)T represent the order of compartments with infection.

We define F and V as follows. The column vector F=βY1+Y2NS,0,0,0,0,0T represents new infections that are introduced into each compartment. The column vector (18) V=σ+τE+μEλ1+τY1+μY1−σEλ2+τY2+μY2−pY1Y2λ1Y1+ρ1L1+pL2Y2ρ2L2μ+τY3Y3−pY2Y3λ2Y2+pL2Y3ρ2L2ρ1+τL1+μL1−pY1L1λ1Y1ρ2+τL2+μL2−pY2L2λ2Y2

represents difference between transfer out of the compartment and the transfer into that compartment that does not result from new infection.

Let F be the Jacobian matrix of F at the disease-free equilibrium, i.e., (19) F=0ββ000000000000000000000000000000000.

Let V be the Jacobian of V at the disease-free equilibrium, i.e., (20) V=σ+τE+μ00000−σλ1+τY1+μ00000−pY1Y2λ1λ2+τY2+μ0−ρ1−pL2Y2ρ200−pY2Y3λ2μ+τY30−pL2Y3ρ20−pY1L1λ100ρ1+τL1+μ000−pY2L2λ200ρ2+τL2+μ.

Let vI denote the sum of all total rates out of the compartment I, i.e., (21) vE=σ+τE+μ

(22) vY1=λ1+τY1+μ

(23) vY2=λ2+τY2+μ

(24) vY3=μ+τY3

(25) vL1=ρ1+τL1+μ

(26) vL2=ρ2+τL2+μ.

Using symbolic computation capabilities of MATLAB, we can calculate V−1, FV−1 and eigenvalues of FV−1. The code is available in the supplementary material. Since only the first row of F is non-trivial, the same is true about FV−1. Thus, FV−1 has only one non-zero eigenvalue given by (27) R0=βσvEvY11+λ1vL2vL1pY1L1ρ1+pY1Y2vL1vL2vY2−pY2L2λ2pL2Y2ρ2.

Lemma 1 R0 ≥ 0. R0 is decreasing in τ. limτ→∞R0 = 0. Specifically, there is τ large enough such that R0 < 1.

Proof 1. Since vY2 = λ2 + τY2 + μ > pY2L2λ2 and vL2 = ρ2 + τL2 + μ > pL2Y2ρ2, all terms in Eq. (27) are non-negative. 2. We get ∂R0∂τE=−R0σ+τE+μ<0. Similarly, ∂R0∂τY1=−R0λ1+τY1+μ<0. Also, since τY2 appears only in the denominator of R0, ∂R0∂τY2<0. It also follows that ∂R0∂τL1<0 because ∂∂τL1pY1L1ρ1+pY1Y2ρ1+τL1+μρ1+τL1+μ<0. Finally, ∂R0∂τL2<0 because ∂∂τL2ρ2+τL2+μρ2+τL2+μλ2+τL2+μ−pY2L2λ2pL2Y2ρ2<0.

More generally, it can also be shown that R0 decreases directly with respect to τ. Since (28) dR0dτ= ∑I∂R0∂τI⋅∂τI∂τ

and τI = cIτ, we have, by previous calculations, dR0dτ<0.

3. This follows directly from Eq. (27).□

Theorem 1 If R0 < 1, then the disease-free equilibrium is globally asymptotically stable. If R0 > 1, the disease-free equilibrium is unstable.

Proof When R0 > 1, the disease-free equilibrium is unstable by van den Driessche & Watmough (2002).

When R0 < 1, the global stability follows from Castillo-Chavez et al. (2002). First, by Eq. (16), S0=Λμ, corresponding to the disease-free equilibrium, is globally asymptotically stable for Eq. (15). Thus, the assumption (H1) in Castillo-Chavez et al. (2002) is satisfied.

Second, let I = (E, Y1, Y2, Y3, L1, L2)T be the vector corresponding to infected compartments. The dynamics of I could thus be described by (29) dIdt=F − VS,I=F−VI−G ^S,I

where F and V are given by Eq. (19) matrix and Eq. (20) matrix and G ^S,I=βY1+Y21−SN,0,0,0,0,0T. Note that F − V is an M-matrix (all the off-diagonal entries are non-negative) and all entries of G ^S,I are non-negative since S ≤ N. Also, F−V=DIF − VS0,0. Thus, the assumption (H2) of Castillo-Chavez et al. (2002) is satisfied. Hence, the disease-free equilibrium (S0, 0) is globally asymptotically stable when R0 < 1.□

Remark.

It follows from Castillo-Chavez et al. (2002) that there is C > 0 such that ∥dIdt∥≤CR0t. Consequently, the time needed the infections to drop below a predetermined level is proportional to (ln(R0))−1.

A.3 Endemic equilibrium

When solving for equilibria of the dynamics, i.e., the constant solutions of Eqs. (8) –(14), we set the left-hand sides to zero and solve the following system of algebraic equations. (30) 0=Λ+τY1Y1+τY2Y2+τEE+τY3Y3+τL1L1+τL2L2−βY1+Y2N+μS

(31) 0=βY1+Y2NS−σ+τE+μE

(32) 0=σE−λ1+τY1+μY1

(33) 0=pY1Y2λ1Y1+ρ1L1+pL2Y2ρ2L2−λ2+τY2+μY2

(34) 0=pY2Y3λ2Y2+pL2Y3ρ2L2−μ+τY3Y3

(35) 0=pY1L1λ1Y1−ρ1+τL1+μL1

(36) 0=pY2L2λ2Y2−ρ2+τL2+μL2.

Let (37) N=S+E+Y1+Y2+Y3+L1+L2.

By adding Eqs. (30) –(36) and solving for N, we get (38) N=Λμ.

By Eq. (32), (39) Y1=kY1E

where kY1=σvY1.

By Eqs. (35) and (39), (40) 0=pY1L1λ1σvY1E−vL1L1

and thus (41) L1=kL1E

where kL1=pY1L1λ1σvY1vL1.

By Eq. (36), (42) L2=pY2L2λ2Y2vL2.

By Eqs. (33), (39), (41), and (42), 0=pY1Y2λ1kY1+ρ1kL1E+pL2Y2ρ2pY2L2λ2Y2vL2−vY2Y2.

Solving for Y2 yields (43) Y2=kY2E

where kY2=pY1Y2λ1kY1+ρ1kL1vL2vY2vL2−pL2Y2ρ2pY2L2λ2

Note that, as in Lemma 1, kY2 > 0. By Eqs. (42) and (43), (44) L2=kL2E

where kL2=pY2L2λ2kY2vL2.

By Eqs. (34), (43), and (44), (45) Y3=kY3E

where (46) kY3=pY2Y3λ2kY2+pL2Y3ρ2kL2vY3.

By Eqs. (31), (43), and (39), (47) S=NvEβkY1+kY2.

Lemma 2 βkY1+kY2=vER0.

Proof (48) βkY1+kY2=βσvY1+βσλ1vL2pY1Y2vL1+pY1L1ρ1vY1vL1vY2vL2−pY2L2λ2pL2Y2ρ2

(49) =βσvY11+λ1vL2pY1Y2vL1+pY1L1ρ1vL1vL2vY2−pY2L2λ2pL2Y2ρ2

(50) =vEβσvEvY11+λ1vL2vL1pY1Y2vL1+pY1L1ρ1vL2vY2−pY2L2λ2pL2Y2ρ2

(51) =vER0.□

Thus, by Lemma 2, Eqs. (47), (37), (39), (43), (45), (41), and (44), we have N=NR0+1+kY1+kY2+kY3+kL1+kL2E.

The results of this section can be summarized in the following theorem.

Theorem 2 The endemic equilibrium E∗=S∗,E∗,Y1∗,Y2∗,Y3∗,L1∗,L2∗ exists and is unique if R0 > 1. It is given by

(52) S∗=ΛμR0,

(53) E∗=Λμ1−1R011+kY1+kY2+kY3+kL1+kL2

(54) I∗=kIE∗,forI∈Y1,Y2,Y3,L1,L2,

where (55) kY1=σvY1

(56) kL1=pY1L1λ1σvY1vL1

(57) kY2=pY1Y2λ1kY1+ρ1kL1vL2vY2vL2−pL2Y2ρ2pY2L2λ2

(58) kL2=pY2L2λ2kY2vL2

(59) kY3=pY2Y3λ2kY2+pL2Y3ρ2kL2vY3.

Appendix B. Model calibration

With the exception of the transmission rate β, the model parameters listed in Table 1 can be found directly in the literature.

The birth rate in Papua New Guinea is 27.2 births per thousand per year (United Nations, 2019). The life expectancy of 65 years (World Bank, 2019). Should the model be used for other countries, we suggest to change these values appropriately since the uncertainty analysis suggest some sensitivity to these values as shown in Fig. 6.

The incubation period, σ−1, after exposure to yaws lasts on average 21 days with a range from 9 to 90 days (Perine et al., 1984; WHO, 2018a). Primary lesions last for 3 to 6 months (Perine et al., 1984). We will assume λ1−1=3 months since this allowed the best fit for our model; larger λ causes larger discrepancies between active and latent yaws cases. To estimate the length of the latent period after primary yaws, we note that secondary yaws occurs one to two months after the primary lesion heals (Marks et al., 2015a). We thus set ρ1−1=1.5 months. All secondary yaws lesions subside in weeks to months (Mitjà, Asiedu & Mabey, 2013) and we will thus assume λ2−1=3 months to be on par with the primary yaws. We note that we are primarily interested in the duration of the infectiousness. The estimated total duration of infectiousness for an untreated yaws patient, including relapses, is of the order of 12 –18 months (Perine et al., 1984). With λ1−1=λ2−1=3 months, this would mean primary yaws, secondary yaws and two to four relapses into secondary yaws.

The second stage of latency ranges from zero to five years, and even up to ten years (Perine et al., 1984; Marks et al., 2015b). Thus, we assume ρ2−1=30 months.

Up to 10% of individuals develop tertiary yaws after five to ten years of untreated infection, but the condition is now extremely rare (Mitjà, Asiedu & Mabey, 2013). We thus set pY2Y3 = 0.0001 and pL2Y2 = 0.9999. With these values, our model estimates the prevalence of the tertiary yaws in the endemic equilibrium (of untreated population) under 0.1% of the total population and slightly under 5% of the number of secondary yaws cases.

About 9–15% of primary yaws cases progress into secondary yaws with the primary lesion still present (Mitjà, Asiedu & Mabey, 2013; Marks et al., 2015b). We thus assumed pY1Y2 = 0.12 for the probability that individual progresses directly from the primary yaws to secondary yaws without any noticeable latent period.

To estimate the value of the transmission rate β, we fitted our model to the baseline data from Lihir Island (Mitjà et al., 2015b). The study indicates that, before the mass treatment trial, 2.4% of the population had active yaws and 18.9% were in the latent stage. The endemic equilibrium of our model given by Theorem 2 can be expressed as a function of β. We used MATLAB’s optimization toolbox to numerically find the value of β so that the endemic equilibrium distribution of S + E, Y1 + Y2 + Y3, and L1 + L2 fits best the empirical values 0.797, 0.024, and 0.189. We got β = 0.016581. With this value, the data from Lihir Island were recovered with an error less than 0.005; most of the error was caused by underestimating active yaws cases.

We note that we ran a number of different scenarios and by allowing a non-zero probability for the individuals with primary or secondary yaws infections to completely recover from the infections and become susceptible, we were able to match data from Lihir Island with the error less than 10−6 (and very similar model outcomes, in particular still 3.5 years to elimination by using TCT and about 20 years to elimination by using TTT).

Appendix C. Global uncertainty and sensitivity analysis

We performed the global uncertainty and sensitivity analysis by the partial rank correlation coefficients (Marino et al., 2008). For every parameter with the exception the treatment rate τI (that does not have any influence at the base prior the beginning of the treatment) and the unknown transmission rate β, we randomly assigned a value from the uniform distribution within the range specified in Table 1. We then tried to fit the transmission rate β to the baseline data while keeping the treatment rate 0. We used the Matlab optimization toolbox as described in the previous section. We then used 1000 sets of the values that could fit to baseline data from Lihir Island (Mitjà et al., 2015b) within an error of 0.005 or less. The actual distribution of the parameter values used in the uncertainty analysis is shown in Fig. 8.

Figure 8 Distribution of the parameter values used in the uncertainty and sensitivity analysis.

The parameters were chosen with ranges as specified in Table 1 such that the model fits the data from Lihir Island (Mitjà et al., 2015b).

For each of these parameter values, we calculated the basic reproduction number, and the time needed for yaws elimination under the TCT and TTT protocols. The resulting histograms are shown in the main body of the manuscript.

Additional Information and Declarations

Competing Interests

Author Contributions

Data Availability

The authors declare there are no competing interests.

Presley Kimball, Jacob Levenson, Amy Moore, Jan Rychtar and Dewey Taylor conceived and designed the experiments, performed the experiments, analyzed the data, prepared figures and/or tables, authored or reviewed drafts of the paper, and approved the final draft.

The following information was supplied regarding data availability:

The MATLAB code for running the numerical ODE solutions is available in the Supplemental Files. The data used in this study is available in Table 1.

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
