# Peer review of "An ODE model of yaws elimination in Lihir Island, Papua New Guinea"

_PeerJ, doi:10.7717/peerj.13018_

## Round 0.1 · original submission · Major Revisions

Thank you very much for submitting your manuscript " An ODE model of yaws elimination in Lihir Island, Papua New Guinea" for consideration at PeerJ. As with all papers reviewed by the journal, your manuscript was reviewed by members of the editorial board and by several independent reviewers. In light of the reviews, we would like to invite the resubmission of a significantly-revised version that fully takes into account the reviewers' comments. More specifically, the reviewers and I have significant concerns about your methodology and the in-depth of your analysis. Details are provided in the reviewers' comments.

We cannot make any decision about publication until we have seen the revised manuscript and your response to the reviewers' comments. Your revised manuscript is also likely to be sent to reviewers for further evaluation.

Reviewer 1 ·

Basic reporting

The authors proposed and analyzed a deterministic model for yaws transmission and control. Yaws was one of the first diseases targeted for eradication in the 1950s. The World Health Organization (WHO) renewed global efforts to eradicate yaws by 2020 in 2012. Despite this, the disease has been one of the most neglected of al the neglected tropical diseases (NTDs). After carefully going through the manuscript I believe the article can be considered for publication if the authors incorporate the following suggestions:

On page 2 (Line 71-75): More detail is required. It is not clear what gap this study is intending to fill. You need to highlight aspects or factors that you incorporated in your framework that makes it unique, reasonably accurate and realistic. I suggest that some of the information which is appearing in the Discussion section about the uniqueness of the proposed model and those in literature should also be in this paragraph.

Experimental design

The proposed model is fine, however the more analysis which leads to more essential results is required.

Validity of the findings

A) The main outcome from this study was to determine threshold conditions for yaws eradication through using active and passive case detection and prompt treatment. In particular, individuals in compartments $E,$ $Y_1$, $L_1$, $Y_2,$ $ L_2$ and $Y_3,$ where assumed to receive treatment. Considering that this disease affects individuals living in poorest communities with little or no access to health how feasible will be it to detect individuals in $E,$ $L_1,$ and $L_2$ under both strategies TTT and TCT?

B) Based on item (A), I suggest that more analysis be done on the proposed model in order to determine different threshold values for disease eradication under different circumstances. After that a comprehensive comparison on the outcomes can be made. For example, one will be left wondering how long it will take to eradicate the disease when only active cases are considered for treatment, since the disease is predominant in poorest communities. By merely comparing the treatment coverage for TCT and TTT one will not be surprised by the outcomes contained in the article.

Additional comments

A) What are the implications of the relapsing individuals on both short and long-term dynamics of the disease?

B) The main results of the manuscript are based on evaluating $R_0$ and one is left wondering the implications of the initial population levels on both short and long-term dynamics of the disease?

Reviewer 2 ·

Basic reporting

--
Review comments for manuscript ID: "66746-v0", entitled "An ODE model of yaws elimination in Lihir Island, Papua New Guinea" of the journal "PeerJ".


--
General:

Kimball et al. conducted a modeling analysis to investigate the transmission dynamics of yaws in Lihir Island, Papua New Guinea. The authors developed a compartmental ODE model of yaws transmission and treatment for these scenarios. The qualitative stability of the model was analyzed and computed for disease-free and endemic equilibria. They calibrated the model and validated its predictions on the data from Lihir Island in Papua New Guinea. They demonstrated that the total targeted treatment (TTT) strategy is efficient in preventing outbreaks. However, due to the presence of asymptomatic latent cases, TTT will not eliminate yaws within a reasonable time frame. To achieve the 2030 eradication target for yaws elimination, the authors suggested that total community treatment (TCT) should be applied instead.

This is a nice and well-conducted modelling study, which addressed an epidemiological problem of timely and public health importance. I consider their method reasonable and sophisticated, and the results are presented logically and consistently. The analyzing outcomes support the main results. Thus, I would recommend it for publishing after the following concerns are considered.


--
Comments:

I suggest the authors give brief epidemiological background on the yaws transmission in the introduction section, like mention how many people get infected or die due to the disease each year in Guinea and globally.

--The method used for some parameter estimation is not very clear. I suggest the authors should explicitly state how they estimate some parameters of the model? To allow others to adopt a similar technique in other settings.

--I appreciate the authors have done a very nice discussion, mainly focusing on the technical part. I would be more appreciative if the author could elaborate on the epidemiology and public health sides.

--Limitations of the study should be clearly stated in the discussion section. Also, I wonder what would be a possible epidemiological or clinical implication of the re-infected on the overall transmission scenarios? Could the authors comment on this based on the proposed model?

--The data-fitting result of the latent yaws using the proposed model is not well fitted? Any explanation for that?

--Did the authors attempt to prove the global asymptotic stability of the endemic equilibrium? If not, I suggest the authors should do that.

Experimental design

See my basic reporting comments

Validity of the findings

See my basic reporting comments

Additional comments

See my basic reporting comments

Reviewer 3 ·

Basic reporting

There is still some room for authors to improve their mathematical language.(for example, "solved the system for DFE and EE", "created the compartmental model")

Experimental design

I have several major concerns in methodology.
First, for the population that progress from Y1 to L1 and from Y1 to Y2, the model used "lambda_1" for both of them. Does it mean that the time that progress from Y1 to L1 is the same as the time that progress from Y1 to Y2? I have similar concern for lambda_2, rho_1 and rho_2. For the use of "rho_2", can I assume that: during the 2nd latent period, on average, it takes the same length of time for them to "relapse into secondary yaws" and "develop into late yaws".

I noticed that values of these parameters were obtained directly from previous studies and authors' assumption except the transmission rate beta. The values of parameter that are estimated by fitting data in reality certainly can improve validity in this study. Unfortunately, this part was not stated clearly in the main part of this manuscript. For example, the type, source, time, location and availability of data should be reported in METHODS. Also, how did you fit the model to data and what is the fitting algorithm applied here should also be reported clearly.

There are some parameters who were assigned values very arbitrarily. For example, (page 3, line 94) the assumed value of tau_L2 is "0.01" just because it is assumed that TTT strategy does not find too many secondary latent case. In fact, "0.01" here is equivalent to say that, on average, it takes 100 months to treat a L2 individual. Is that something assumed by authors? Will the results of this study keep unchanged if we assign another number to this parameter?

Validity of the findings

no comment.

---

## Round 0.2 · Minor Revisions

The two comments of reviewer #2 should be thoroughly addressed before the manuscript can be deemed suitable for publication.

Reviewer 1 ·

Basic reporting

I happy with the changes made by the authors. I believe a number of issues that were not clear in the original draft are now clear.

Experimental design

More clarity has been provided by the authors regarding their model formulation.

Validity of the findings

Additional results have been added as well as additional explanations.

Additional comments

Overall, I am happy with the revised manuscript.

Reviewer 2 ·

Basic reporting

--
Review comments for manuscript ID: "66746-v1", entitled "An ODE model of yaws elimination in Lihir Island, Papua New Guinea" of the journal "PeerJ".
* * *
Comments:
I appreciate that the authors have carefully addressed all the issues raised by the reviewers that might improve the manuscript from its initial draft. The paper has improved significantly with well-conducted modelling analyses, which addressed yaws epidemiological problems in Lihir Island, Papua New Guinea. The method was reasonable and sound, and the results were presented logically and consistently. The analyzing outcomes support the main results with biologically and clinically appropriate settings.
Thus, after the following comments are considered, I recommend publishing this work in the prestigious journal of "PeerJ".


--I suggest the authors estimate/plot the confidence interval in figure 3, which could give a more reasonable result.

--Regarding the proof of the global asymptotic stability of the endemic equilibrium, the authors claimed that they did not succeed. Though not necessary in the current work, the authors can use a similar approach as employed in the following works:
1) La Salle JP. The stability of dynamical systems. Society for Industrial and Applied Mathematics; 1976.
2) Yang C, Wang X, Gao D, Wang J. Impact of awareness programs on cholera dynamics: two modeling approaches. Bulletin of mathematical biology. 2017;79(9):2109-31.
3) Roop-O P, Chinviriyasit W, Chinviriyasit S. The effect of incidence function in backward bifurcation for malaria model with temporary immunity. Mathematical biosciences. 2015;265:47-64.

--Finally, I also suggest rigorous proofread of the manuscript.

Experimental design

see, basic reporting comments

Validity of the findings

see, basic reporting comments

Additional comments

None

---

## Round 0.3 · accepted · Accept

Thank you for thoroughly addressing the reviewers' comments. It has been a long process, but we hope that you are happy with the improved quality of the manuscript. Thank you for your support of PeerJ

Reviewer 2 ·

Basic reporting
* * *
R2: Review comments for manuscript ID: “peerj-66746”, entitled “An ODE model of yaws elimination in Lihir Island, Papua New Guinea” of the journal “PeerJ”.
* * *
General:
I appreciate that the authors have carefully addressed all the issues raised by the reviewers that might improve the manuscript from its initial draft. I think the paper is recommended for publication after consideration of some minor comments below, which needs no further revision.
* * *
Comments:
--Regarding the GAS proof using the LaSalle Invariant Principle, since the authors did not succeed with the proof, I think there is no need in defining the Lyapunov function (in the added paragraph). Numerical simulations would be sufficient. However, the authors can still cite LaSalle and Yang et al.’s works.

Experimental design

See, basic reporting comments.

Validity of the findings

See, basic reporting comments.

Additional comments

See, basic reporting comments.